# Entropic Dynamics in a Theoretical Framework for Biosystems

**DOI:** 10.3390/e25030528

**Published:** 2023-03-18

**Authors:** Richard L. Summers

**Affiliations:** Department of Physiology & Biophysics, University of Mississippi Medical Center, Jackson, MS 39216, USA; rsummers@umc.edu; Tel.: +1-01-260-1568

**Keywords:** entropic dynamics, biosystems, Kullback principle of minimum information discrimination, biocontinuum, information geometry

## Abstract

Central to an understanding of the physical nature of biosystems is an apprehension of their ability to control entropy dynamics in their environment. To achieve ongoing stability and survival, living systems must adaptively respond to incoming information signals concerning matter and energy perturbations in their biological continuum (biocontinuum). Entropy dynamics for the living system are then determined by the natural drive for reconciliation of these information divergences in the context of the constraints formed by the geometry of the biocontinuum information space. The configuration of this information geometry is determined by the inherent biological structure, processes and adaptive controls that are necessary for the stable functioning of the organism. The trajectory of this adaptive reconciliation process can be described by an information-theoretic formulation of the living system’s procedure for actionable knowledge acquisition that incorporates the axiomatic inference of the Kullback principle of minimum information discrimination (a derivative of Jaynes’ principle of maximal entropy). Utilizing relative information for entropic inference provides for the incorporation of a background of the adaptive constraints in biosystems within the operations of Fisher biologic replicator dynamics. This mathematical expression for entropic dynamics within the biocontinuum may then serve as a theoretical framework for the general analysis of biological phenomena.

## 1. Introduction

In his famous 1943 lecture entitled *What is Life?*, Nobel-prize-winning physicist Erwin Schrödinger proposed that a true understanding of the physical nature of living systems first requires an apprehension of their ability to control entropy dynamics in their environment [1]. However, when the reductionist approach of classical physics began to be applied to investigate the complexity of living organisms, it became evident that these traditional methods were inadequate for completely describing their functioning. Nevertheless, it is still important to be able to derive the global emergent behaviors inherent in biological systems from fundamental physical processes.

An alternative analytic approach that is congruent with the physical sciences is critical for the future advancement of the life sciences. Ariel Caticha, Carlo Cafaro and others have proposed that all phenomenal dynamics are not really based on laws of science but rather arise from rules for processing information about nature [2,3]. In this perspective, the emergent axioms of biological complexity are not based in physical laws but rather are founded in the procedures for inference in the context of controlling constraints and guidance of the overall system directives. This frame of reference that is just now being discovered creates a systems level hierarchy of governance beyond basic physical laws. 

Significant contributions by Ilya Prigogine and Harold Morowitz in the field of nonequilibirum thermodynamics have demonstrated that energy flows through attractor systems have a unique capacity to create localized states of order as seen in open living systems [4,5]. Procedures that utilize information, such as those employed by Maxwell’s demon and Feynman’s ratchet, were also considered by physicists such as Leo Sziliard and Léon Brillouin to overcome the entropic tendencies toward disorder and were thought to be critical for the emergence of organized living systems [6,7]. 

The physical notions of entropy have historically been thought to be closely aligned with Shannon’s conception of information [8,9]. However, initially, there was no definitive relationship between entropy and information beyond their functions as analogous measures of global system uncertainty. A fundamental connection was finally revealed when E. T. Jaynes demonstrated a proof of the second law of thermodynamics that involved a transition of the microscopically based Boltzman formula to the macroscopic state using Liouville’s theorem [10]. This determination led to the consideration that a similar linkage of our modern understanding of physical entropy with the well-established Shannon communication process as the biologic currency of information could be the pathway for the realization of Schrödinger’s insight [9].

Sara I. Walker and Paul Davies have posited that the algorithmic processing of information is the most essential function of living systems [11]. It is really the ability of these biosystems to acquire information signals concerning matter and energy perturbations in their biological continuum (biocontinuum) and translate that information into adaptive actions that are critical to their stability and survival. More recently, it has been suggested that this process of entropic information acquisition and conversion to actionable knowledge and meaning is the key defining characteristic of living organisms [9]. Carlo Rovelli further proposes that a physical grounding of this meaning for living systems can be achieved by combining the notions of relative information with the stability and survival mechanics of Darwinian adaptation [12]. 

In this paper, an inference framework was derived describing the natural axiomatic procedures used by living systems for processing information concerning the physical state of their biocontinuum. The objective of these innate operations is to translate the acquired entropic information of the organism’s biocontinuum into actionable knowledge for adaptive reconciliation toward system stability. This derived framework incorporates the methods of entropic dynamics into the natural biologic processes of Darwinian replicator dynamics for a cohesive and coherent global approach to understanding biologic organizational complexity and the localized control of entropy in living systems. 

## 2. Materials and Methods

### 2.1. Entropic Dynamics

Caticha and Cafaro have suggested that phenomenal dynamics are more fundamentally predicted by principles of inference rather than any derived scientific laws [2,3]. Entropic dynamics is a procedural framework for the derivation of system dynamics from the standard probabilistic rules for inference and the processing of information based on Jaynes’ principle of maximal entropy [10]. In this procedure, entropy provides the driving force for system changes and Jaynes’ entropic methods of inference delineate the characteristics of those changes when subject to the constraints inherent in the system’s structural and dynamic functioning. Systems progress from an initial probable state to a new most probable state depending on changes in information about the state of the system [10]. This methodology provides a unique perspective in the analysis of phenomenal dynamics. Using this information-theoretic approach, physicists have been able to rederive many of the standard models in the physical sciences [13,14,15]. However, for biosystems the constraints are very different, being highly mutable and uniquely adaptive toward the intrinsic system objectives of stability and sustainability. Therefore, a relational form of Jaynesian entropic inference that utilizes Kullback–Leibler information divergence as relative information is adopted to account for this biosystem context [16]. The unique framework devised herein incorporates the methods of entropic dynamics in the processing of Fisher’s Darwinian adaptive replicator functions for predicting the trajectory of phenomenal states in living systems. This integration provides for a more cohesive and coherent global approach to understanding biologic phenomena and dynamics in the determination of organizational complexity and localized control of entropy in living systems. 

### 2.2. Kullback Principle of Minimum Information Discrimination

The Jaynesian procedure for inference is known as the principle of maximal entropy [10,17]. This principle posits that most inferences are made based on incomplete information and that they should be drawn from that probability distribution that has the maximum entropy permitted by the available information. In other words, when there is incoming information concerning a system perturbation, then the inferred update to the probability distribution concerning the new state of the system should be minimal in its discrepancy from the original distribution with as small a gain in information as possible. This inferred distribution is considered to represent the most conservative assignment of values that only draws conclusions substantiated by the known information and constraints. The information dynamics for this reconciliation of state are considered the mechanism that drives a change in a conclusion during the inference process. This inference process is also considered the basis of the entropic drive for the physical dispersal of matter and energy to maximum disorder based on the logical statistical mechanisms of chance variations. In fact, this statistically derived entropic drive is the same as that used in traditional mathematical descriptions of entropy transitions in physical phenomena. Hence, all systems naturally evolve to reconcile the introduction of new conditions in a way that maximizes their entropy and minimizes their gain in order or information.

The Jaynesian methodology also connects any intentional inquiry concerning physical phenomena with information theory and the process of inductive logic and inferential reasoning. Intentional inquiry as an active request for information invariably produces some uncertainty in the results represented as probability distributions. The Jaynes method allows for inferences to conclusions based on the most likely probability distribution given the limited information [10,17,18,19]. The MaxENT procedure of Jaynes has been successfully used as an alternative method to derive statistical mechanics and has been applied in the sophisticated analysis of a variety of physical phenomena [13,14].

A method similar to Jaynes’ principle of maximal entropy was provided by Solomon Kullback for the case of relative information divergences within a system and is known as the principle of minimum information discrimination [18,19]. This principle considers the inference procedure from the perspective of the evolving relative entropic information differences across the continuum of the system. Entropy is then maximized globally by minimizing the entropic information differences across the system continuum subject to any localized constraints. Because all known phenomena in material reality are fundamentally based on relational interactions, Carlo Rovelli considers that the relative nature of the Kullback–Leibler information divergence metric makes it the true physical version of Shannon-type information measurement [12]. Hence, in the case where new relative information is acquired, a new distribution of the probability should be inferred which minimizes the discrimination from the original distribution across the entire system continuum as much as possible. In this way, the new data produces as small an information gain as is possible and optimizes the veracity of induction. This principle of minimum information discrimination then becomes the guiding principle employed by the driving entropic force for directed change toward a conclusion during the inference process [20]. According to Kullback, information acquired from any observations can also be considered relative to expectation values over the probability space. Such expectant relative information measures (Kullback–Leibler divergence) denote the difference between prior information and any new information as an analog to a Bayesian update processing. 

Because of the important implications of these ideas, J. E. Shore and R. W. Johnson provided a detailed axiomatic examination of Jaynes’s principle of maximum entropy and Kullback’s principle of minimum discrimination information [9,21]. They concluded that the procedures of these principles are logically consistent methods of inference from new information. The axioms employed were based on fundamental first principles requiring a consistent probability model of inductive inference and reliable results regardless of the solution pathway. The analytic approach of Shore and Johnson did not depend on intuitive arguments or rely on the properties of entropy and Kullback–Leibler divergence as measures of information but rather they created basic axioms containing certain desired properties for the inference methods. Their proofs based on these consistency axioms demonstrated there is only one probability distribution solution satisfying the introduction of new information constraints. The axioms of Shore and Johnson included:Uniqueness: the result of the inference should be unique.Invariance: the choice of a coordinate system should not matter.System independence: it should not matter whether one accounts for independent information about independent systems separately in terms of different densities or together in terms of a joint density.Subset independence: it should not matter whether one treats an independent subset of system states in terms of a separate conditional density or in terms of the full system density.

### 2.3. Biological Continuum (Biocontinuum)

The contextual mathematical and geometric construct used in the current analysis of action within the living system’s internal and external milieu is its biologic continuum, termed the biocontinuum [9,20]. A continuum is a continuous nonspatial whole or succession of states. This structure provides for the seamless union of entities into an amalgamated ensemble such as the framework that joins space and time into a single geometric structure. The biocontinuum is defined as a coherent information state space that includes everything having a potential information interchange with the life system processes. So that biocontinuum space comprises all possible energy and material exchanges as well as any informational communiques originating from within or external to the usual considered boundaries of the organism. As a continuum, there is no distinction between the living system and its embedded environment within the experiential realm of the life processes, including the organism’s own state. The inseparability of the open living system from its environment is a common notion in many modern biological theoretical constructs [22,23]. As defined by Gregory Newby, an information space is a set of concepts and the relations between them that are contained in an informational system [24]. Therefore, the biocontinuum can be considered as a space containing a set of systematically and logically interconnected pieces of information as a coherent whole and therefore defines the information state of the organism and its environment. Such a space also describes the range of all possible values and relationships an entity can have under a given set of rules and conditions. 

The information in the biocontinuum space described here takes the form of Shannon information represented as the probability spectrum of possible state conditions in which the points in a manifold of a two-dimensional geometry can represent the mean µ and variance σ of the probability distribution [25]. This formulation allows for a geometric representation of the biocontinuum in which the contours of the space provide a natural trajectory of action dynamics as driven by the living system’s active adaptation to changing conditions. The dimensions of the geometrical/topological measures of physical phenomena can be specified in terms of the number of points in a coordinate system. It is within this dimensional construct that states are differentiated and become materially known to the organism. Therefore, the unique dimensionality of the biocontinuum platform is determined by the sensory discerning and measurement capacity of the observing system as a signature composed of informational metrics. Even at the cellular level, chemo-tactile facilities exist to distinguish sequential time, acceleration and some rudimentary form of spatial dimensionality [9].

### 2.4. Information Geometry

Geometric structures are often used to define and analyze the dynamics of physical systems. Caticha suggests that the geometry of space and time is simply a macroscopic manifestation of an underlying statistical structure [13,14,26]. Frieden has devised a similar methodology using Fisher information metrics [15]. The dynamics of phenomena evolving in space and time can then be described by the geometry of this statistical structure. Information geometry, as developed by Amari in the 1980s, is the application of the methods of conventional Riemann geometry to the analysis of the differential geometric structure of evolving probability distributions as statistical manifolds [25]. In this method, each information point in a geometry of an n-dimensional manifold space can be associated with the defining parameters of some model describing the probability distribution. In this form, the degree to which one probability distribution point can be distinguished from another in the geometric information space is through a measure of the distance between the points. Utilizing the Kullback–Leibler information divergence and principle of minimum information discrimination localizes the smallest distinguishable distance and quantifies this distinction in units of uncertainty. Dissipative entropy dynamics then follow the trajectory of the system, which moves continuously and irreversibly along free energy gradients in a geodesic space of probable states. Such mathematical constructs can also be used to understand information exchanges along functional gradients within living systems. The advantage of utilizing this geometrical approach is its capacity for a larger perspective and fundamental analysis where the focus is on the most probable trajectory as driven by entropy and guided by logical inference principles and not based on proposed laws of physics or any action principle. 

### 2.5. Replicator Dynamics

In the current formulation, the mechanics of the replicator equations form the underlying engine for using entropic dynamics and inference in biosystems in the derived framework. These equations were introduced by R. A. Fisher in the 1930s to model the fundamentals of Darwin’s idea of natural selection through the process of survival based on an organism’s fitness and stability [27,28]. The basic theorem of Fisher can be stated as “The rate of increase in fitness of any organism at any time is equal to its variance in fitness at that time.” This theorem captures the central idea of the natural selection process in that the population types with greater than average fitness should increase in their proportion in the population. In other words, the relative growth rate of each population constituent type should be proportional to the difference between the fitness of the type and the mean fitness in the overall population. Fisher and Haldane further used these mathematical models to create a synthesis between Mendelian genetics and Darwinian evolution [27,29]. This approach also provides a basis for a grounding for the meaning of information using this Darwinian fitness calculus [12]. 

The mechanics of the replicator mathematical expressions also provide the logical essence for natural selection driven by competitive dynamics. Through these mechanics and the resulting dynamics, the relative propensity of states of a replicating entity is determined by the difference in its ability to endure (fitness) as compared to other competing entities within its proximity. From these replicator dynamics, there is the natural emergence of an entity that is more adaptive and robust within the conditions of the environment. The procedures that select and propagate system sustainability over time are logically those also geared for that specific objective and therefore are the ones that naturally persist. Since the fundamental mechanism used in any adaptive procedure is the perception-action process, the replicator expression incorporates these mechanics in its execution of the natural selection process. 

Since its inception, the replicator equation has become one of the most important dynamic models in biology, ecology, evolutionary studies and even such diverse fields as economics and sociology. The current modern general form of the replicator equation was structured by Taylor and Jonkers in the 1970s [30] as:$$\dot{x_{i}}=x_{i}\left[f_{i}\left(x\right)-\varphi \left(x\right)\right]$$
$$\varphi \left(x\right)=\overset{n}{\underset{i}{\sum }}x_{i}f_{i}\left(x\right)$$
where:

$x_{i}$ is the proportion of type *i* in the population with the type being any principal attribute category of determined variation and *x* is the rate of change. $f_{i}\left(x\right)$ is the fitness of each type i in the population with fitness being a survival likelihood characteristic in the context of the environment. φ(x) is the average population fitness as determined by the weighted average of the fitness of the overall population.

As an iterative dynamic with time, it is easy to see how the mechanics of this equation can change the constituent states of the population in a way that results in the optimal mean state with the greatest chance for survival in the environment. Subsequently, there is a continuous cyclic renewal and propagation of the overall state of the biocontinuum. Because of the significance of this archetype to an understanding of the overall framework for biological dynamics, it is important to understand its derivation from logical first principles and the current incorporation of the Kullback principle of minimum information discrimination as demonstrated in the following section. 

## 3. Results

The theoretical framework derived for the analysis of biological systems was structured in a way to broadly describe the homeorhetic functioning of these systems based on first principles including information processing for the adaptive reconciliation of entropy divergences within the biocontinuum in order to maintain stability and survival [9]. These divergences provide the entropic drive for these system dynamics as determined by the inference procedure of the Kullback principle of minimum information discrimination and in the context of the inherent constraints of the biology. 

### 3.1. Derivation of Equations of Entropic Dynamics for the Biosystem

While the basic driving forces of entropic dynamics are the same for biology as they are for any physical system, the utilization of a particular form of Jaynesian entropic inference (Kullback’s principle of minimum information discrimination) and the adaptive constraints of living systems are unique. Those distinctive features are incorporated into a mathematical framework for the analysis of biosystem dynamics. The framework mechanics of the derived mathematical expressions are also founded on the logical essence of the biological natural selection process as modeled by Fisher’s equations of replicator dynamics [27]. For a general population (*P*) of organisms with variations of type (*i*), the Lotka–Volterra derivative of the replicator equation describes a very general rule for how these population numbers can change with time and is defined as [31]:$$\frac{dP_{i}}{dt}=f_{i}\left(P\right)P_{i}$$

However, rather than considering the probability distribution of a population, the expression can also be employed to describe the dynamics of possible states for the biocontinuum of a single organism. As an iterative dynamic with time, there is a continuous cyclic renewal and propagation of the state of that biocontinuum for the singular living system. In this expression, the relative sustainability of the various possible ith states for an organism is determined by the difference in the state’s ability to endure (fitness: $f_{i}\left(P\right)$) compared to other possibles and serves as a prescribed system constraint. The full range of these functions$\;f_{i}$ determines the fitness landscape for the biocontinuum. Therefore, there is the natural emergence of an organism that is most adapted to the conditions of the biocontinuum with the greatest probability of survival.

In systems control theory, the fitness function is also called an adaptation evaluation function and is used in relation to the error function which determines the difference between actual and *a priori* predicted solutions [32]. System adaptation for minimization of error also minimizes the system’s internal energy as information that is gained is assimilated into the system’s structure and function. Using the systems control approach to fitness evaluation of the single organism allows for integrating dynamic models of the constraints of very complex living system homeorhetic and adaptive functioning into the dynamics.

The Lotka–Volterra expression is made more practical by a normalization of the probabilities of all possible states by letting $p_{i}$ be defined as the probability fraction of the ith state within the entire spectrum of possibles (see below). In this configuration, the values for p are also considered as probabilities where$\;p_{i}$ is the probability that a randomly chosen constituent of the possible states is of the *i*th type.
$$p_{i}=\left(\frac{P_{i}}{\sum_{i}^{n}P_{i}}\right)$$

These values for $p_{i}$ are between 0 and 1 and add up to 1. In this mathematical framework and by the quotient rule of calculus for derivatives, the Lotka–Volterra equation becomes the usual form of the replicator equation:$$\frac{dp_{i}}{dt}={\dot{p}}_{i}=(f_{i}\left(P\right)-\langle f(P)\rangle )p_{i}$$
where: 

〈f(P)〉 is the mean fitness of all the possible system states. 

Additionally, if we consider that each fitness depends on the fraction of each possible state, then the replicator equation simplifies to a more statistically usable form for information theory as:$$\frac{dp_{i}}{dt}=(f_{i}\left(p\right)-\langle f(p)\rangle )p_{i}$$

This expression then determines the change in the probability fraction of a possible biocontinuum state at a rate proportional to the fitness of that state minus the mean fitness. With normalized probability distributions of the possibles, the derived state expression will include information probability and knowledge uncertainty based on Shannon information theory. This information state is defined as the amount of average information required to determine the probability spectrum of possible states. It is also a measure of the uncertainty an observer has about the system state. This measure is the summation of the surprisals and is defined as:$$S\left(p\right)=-\overset{n}{\underset{i}{\sum }}p_{i}ln\left(p_{i}\right)$$

Incorporating these information theory metrics into the replicator expression then allows for the analysis of entropic dynamics in living systems. This framework of uncertainty in the differentiation of the “signals” of information concerning the state of the biocontinuum environment is also consistent with known biological knowledge acquisition processes as the system gains information and adapts to reflect the context of its space. Therefore, the rate of change in this information gain and uncertainty loss (the entropic dynamics) for the living system is given by the equations:$$\dot{S}=-\overset{n}{\underset{i}{\sum }}\dot{p_{i}}ln\left(p_{i}\right)$$
$$\dot{p_{i}}=(f_{i}\left(P\right)-\langle f(P)\rangle )p_{i}$$
$$\dot{S}=-\overset{n}{\underset{i}{\sum }}(f_{i}\left(P\right)-\langle f(P)\rangle )p_{i}ln\left(p_{i}\right)$$

From this expression, it appears that the information gain is mainly dependent on the differential assessment of the relative fitness of the organism to the biocontinuum condition associated with that information, and that fitness is determined by the integrity and robustness of the inherent complex homeorhetic processes and functioning of the living organism. This is because the structural constraints and entropic dynamics of the living system are actively adapted toward the objective of stability and a sustained existence based on this assessment.

In the dynamics of any system, there is a continuous transition from the current global state to an optimal or goal state. This transition requires a reconciliation of divergent conditions toward that optimal state across the whole of the system continuum. Physical phenomena within such systems are fundamentally based on relational interactions and described by the relative information concerning those interactions. Divergences in the entropic information between the current and desired states are best measured by the Kullback–Leibler divergence metric for relative information. The Kullback–Leibler information divergence of P from Q (denoted *D_K_*_L_(Q||P) or I(*q,p*)) is where P is the prior or current probability distribution of types and Q is the distribution that is the optimal end state. By incorporating this measure into the functioning of the replicator expression, the use of prior information within the living system regarding the state of the biocontinuum is possible using a Bayesian inference approach. The Bayesian updating during the iterative procedure with time also accounts for the Landauer erasure of information required for balancing thermodynamic entropy [33]. The information differential between *q* and *p* at any point in the dynamic transition is the remaining information to be learned. The equation for the Kullback–Leibler information divergence is then given by:$$I\left(q,p\right)=D_{KL}(q|p)=\sum_{i}^{n}ln\left(\frac{q_{i}}{p_{i}}\right)q_{i}=\sum_{i}^{n}(ln\;\left(q_{i}\right)-ln\;\left(p_{i}\right))\;q_{i}\;$$For I(q,p) where q is a target goal state with a fixed probability distribution and only p is time dependent then:$$\frac{d}{dt}\sum_{i}^{n}ln\;\left(q_{i}\right)q_{i}=0\;\mathrm{Since}\;q_{i}\mathrm{is}a\mathrm{fixed}\mathrm{quantity}$$
and
$$\frac{d}{dt}I\left(q,p\right)=-\frac{d}{dt}\sum_{i}^{n}ln\;\left(p_{i}\right)q_{i}=-\sum_{i}^{n}\left(\frac{\dot{p_{i}}}{p_{i}}\right)q_{i}$$
where $\dot{p_{i}}\;$is the rate of change of the probability $p_{i}$ and is defined by the replicator equation as: $$\dot{p_{i}}=(f_{i}\left(P\right)-\langle f(P)\rangle )p_{i}$$Substituting this expression into our derivative equation results in:$$\frac{d}{dt}I\left(q,p\right)=-\overset{n}{\underset{i}{\sum }}(f_{i}\left(P\right)-\langle f(P)\rangle )q_{i}$$Since the probability $q_{i}$ sums to one, the equation becomes:$$\frac{d}{dt}I\left(q,p\right)=f\left(P\right)-\sum_{i}^{n}f_{i}\left(P\right)q_{i}\;\;\sum_{i}^{n}f_{i}\left(P\right)(p_{i}-q_{i})$$
where f(P) demarcated in the biocontinuum is the same for $p_{i}\mathrm{and}\;q_{i}$

If the potential information (*I*) as the Kullback–Leibler divergence of the biocontinuum is defined by:$$I\left(q,p\right)=\sum_{i}^{n}ln\left(\frac{q_{i}}{p_{i}}\right)q_{i}=\sum_{i}^{n}(ln\;\left(q_{i}\right)-ln\;\left(p_{i}\right))\;q_{i}\;\;$$

And the kinetic information defined as the changing of the Kullback–Leibler divergence during the procedure of information being assimilated by the adaptive processes of the living system is described by:$$\frac{d}{dt}I\left(q,p\right)=-\sum_{i}^{n}(f_{i}\left(P\right)-\langle f(P)\rangle )q_{i}=\sum_{i}^{n}f_{i}\left(P\right)(p_{i}-q_{i})$$

Then the ***ACTION*** measure for all i elements is:$$ACTION=\int^{\;}\left[\sum_{i}^{n}f_{i}\left(P\right)(p_{i}-q_{i})-\sum_{i}^{n}ln\left(\frac{q_{i}}{p_{i}}\right)q_{i}\;\right]$$
as the integral summation over time of the Lagrangian integrand which is the difference between the kinetic and potentials at each phase of the change transition. These differences naturally act as a variational principle to determine the trajectory in a dynamical system that possesses some particular extremum characteristic. The information divergence also serves as an impetus for change with equivalency to entropic forces as determined by Jaynes [10]. This expression determines the overall evolution and trajectory of the living system as it moves to its final state. However, this unique action principle and function for living systems is not fundamental but arises from the more basic physical dynamics of entropic inference subject to the constraints of the organism’s structure and processes as they are geared for stability and survival. While the inherent dynamics, fitness function and adapting constraints of the replicator equation naturally reduce the system’s information divergences, the broader axiomatic behavior of this process is grounded in the entropic drive of Kullback’s principle of minimum discrimination information.

### 3.2. Information Geometry of the Biological Continuum (Biocontinuum)

In applying the methods of information geometry to the constructed framework for biosystems, a geometric landscape emerges depicting the natural gradient flow of the entropy driven dynamics of biological functioning (example construct in Figure 1) [34]. The contours of that landscape are determined by the intrinsic biosystem constraints including active adaptation processes toward a state of greatest fitness for stability and survival. In this process, the divergence between information states within the biocontinuum is naturally reconciled by the inference process of Kullback’s principle of minimum discrimination information. 

## 4. Discussion

The key to comprehending the unique physical nature of living systems resides in first understanding their local control of entropy dynamics [1]. In fact, the process of adaptive reconciliation of entropic information perturbations within their biocontinuum is most critical to the stability and survival of all living organisms [9]. Entropic dynamics as espoused by Caticha and Cafaro provides a potential physical framework for deriving the dynamics of living systems in this congruency process [9,10]. The special conditions for entropic dynamics within the biocontinuum may then serve as a theoretical framework for the general analysis of biological phenomena.

In this paper, an inference framework and process based on the principles of entropic dynamics were derived that describes the natural axiomatic procedure used by living systems for processing information concerning the physical state of their biocontinuum. The mathematical derivation of these concepts is highly informed by the approach of Harper and Baez in their analysis of evolutionary dynamics [35,36,37]. System entropic information defined as Kullback–Leibler information divergence drives the active responses of the living system as subjected to its variable constraints and organizational structure for inherent homeorhetic activities. The objective of these innate operations is to translate the acquired entropic information into actionable knowledge for adaptive reconciliation toward system stability. The utilization of this relative information and the Kullback’s principle of minimum discrimination information as a variant method of Jaynesian inference provides for the inclusion of the context of an adapting biosystem. By incorporating the methods of entropic dynamics in the processing of Fisher’s Darwinian adaptive replicator functions, this framework can be used for ascertaining the phenomenal dynamics of living systems. Furthermore, Kullback’s principle of minimum discrimination information becomes the foundation of a biologic action principle for the reconciliation of information divergence across the biocontinuum. Interestingly, the Lyapunov vector signature of stability for these dynamics can also serve as a quantitative metric for biosemiotic meaning of the information that is grounded in system stability with existential significance [38,39]. 

A linkage between the living organism’s relative environmental information and its system stability has been previously described by Friston based on fundamental physical principles [40]. He suggests that these living systems maintain their non-equilibrium steady state and restrict their degrees of freedom by actively minimizing the variational free energy of their internal states. This free energy principle is closely related to Varela’s notion of autopoiesis but extends the concept by proposing a general functional control mechanism with the partitioning of external and internal states using a Markov blanket and a representational model for predictions of states. Friston’s free energy principle as an action principle is widely recognized as a salient advance in our understanding of biological systems and is generally consistent with the concepts proposed in this paper. However, from the perspective of entropic dynamics, the free energy action principle is not fundamental since the inference process for action that is attributed to originate from internal representational models really arises from the more basic physical dynamics of a global entropic inference as subjected to the constraints of the organism’s natural homeorhetic processes. This differentiating perspective is supported by the work of Skarda and Freeman in experimental neurodynamics indicating that biological systems deconstruct information about the biocontinuum as a primary whole rather than constructing representations [41,42]. The complex structure of the living then results from the internal integration of the entropic information signals into a perceptual whole. 

The work is also informed by the important ideas of Vanchurin et al. regarding the maximum entropy principle as applied to learning and evolution [43]. 

In the framework of entropic dynamics for biosystems presented in this paper, divergences from steady state within the integrated whole of the organism’s biocontinuum are differentiated and reconciled by entropic inference principles through the deconstruction of information states without the need for representational models [9]. This view is consistent with the modern epistemological notion that informational knowledge acquisition is about determining relationships rather than representing the substance, essence or condition of things [12]. Any construction of representational models for describing the controlling constraints of homeorhesis and fitness evaluation are simply analytic tools and not independent properties of living systems. Such a practical application of the methodology in an established biosystem model was presented in a previous publication as a proof of concept [44]. The subtle difference in this more fundamental, nonrepresentational methodology for analyzing the phenomenal dynamics in living systems is that it provides for a new and more direct understanding of the natural emergence of these systems [9].

Information processing and Bayesian updating have previously been considered critical aspects of biological evolution [45,46,47,48,49]. Entropic dynamics as a methodology carries the promise of a foundational approach to understanding physical phenomena. However, it should be recognized that entropy as a measure of energy quality is calculated in terms of uncertainty metrics. This assessment implies a central role for an adjudicating observer or at least some form of the knowledge acquisition process from which this uncertainty arises. Such a process is naturally inherent in biological system functioning as critical to an organism’s survival. Likewise, the process of inference is in part rooted in the logical reasoning procedure of living organisms. Since entropic dynamics has been so successful in describing many such physical phenomena from a foundational position, it is possible that a generalized methodology that further incorporates the entropy-driven knowledge acquisition process of living systems could provide a more comprehensive approach to all common scientific inquiry.

## Figures and Tables

**Figure 1 entropy-25-00528-f001:**
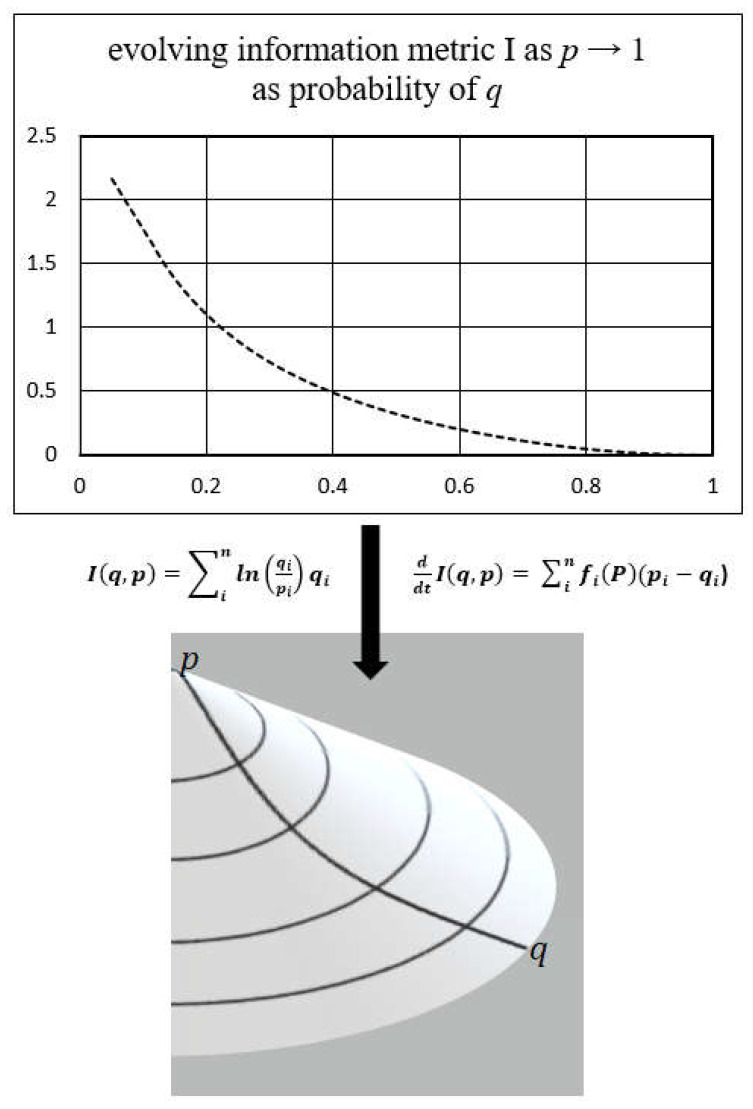
This graphic below depicts an example of the natural evolution $\frac{d}{dt}I\left(q,p\right)\;$of the Kullback-Leibler information divergence I(q,p)  as an action gradient driven by the geometry of the biocontinuum created by the system processes and structure, replicator dynamics, fitness function and the entropic procedure of the Kullback principle of minimization of information. This information metric exists as a probability spectrum (i) of uncertainty in the biocontinuum information space of the biologic system process. As new information is assimilated as knowledge into the adapting system, the prior state p emerges as a new target state q along the gradients of the biocontinuum information space.

## Data Availability

Not applicable.

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
