# Peer review of "Entropic Dynamics in a Theoretical Framework for Biosystems"

_entropy, 2023, doi:10.3390/e25030528_

Round 1

Reviewer 1 Report

Comments: In this paper, the author describes the notion that the dynamics of evolving phenomena in space and time can be explained by the geometry of statistical structures for the exchange of entropic information. Entropic dynamics follows the trajectory of a system that continuously and irreversibly moves along an entropic gradient in the geodesic space of possible states. From this point of view, the author argues that the dynamics of the biocontinuum can be understood through understanding the trajectory. Additionally, he mentions that the change over time of entropy information is characterized as an action.

I find the idea of the connection between entropy and the biocontinuum that the author claims to be rather interesting. However, I believe that these conceptual descriptions alone are not sufficient for this paper to be published. I strongly suggest that the author applys the ideas to specific biosystems and present statistical results through mathematical or physical analysis.

Author Response

Comments: In this paper, the author describes the notion that the dynamics of evolving phenomena in space and time can be explained by the geometry of statistical structures for the exchange of entropic information. Entropic dynamics follows the trajectory of a system that continuously and irreversibly moves along an entropic gradient in the geodesic space of possible states. From this point of view, the author argues that the dynamics of the biocontinuum can be understood through understanding the trajectory. Additionally, he mentions that the change over time of entropy information is characterized as an action.

I find the idea of the connection between entropy and the biocontinuum that the author claims to be rather interesting. However, I believe that these conceptual descriptions alone are not sufficient for this paper to be published. I strongly suggest that the author applys the ideas to specific biosystems and present statistical results through mathematical or physical analysis.

I appreciate the reviewer's comments and interest.  I completely agree with the idea that the method needs to be practically applied.  Such an application was provided in a previous publication in the journal Biosemiotics and a copy of that paper is uploaded for your purview.  A reference to that paper is now included in the manuscript and references. 

Reviewer 2 Report

I enjoyed reading this scholarly and convincing account of entropy dynamics in relation to biological self organisation. I thought the background and review of the theoretical heritage was very nice. On the other hand, some of the references are a bit dated and a bit colloquial. I mention this because I think you are in a very good position to join the dots in contemporary thinking about self organisation, density dynamics, the free energy principle and information geometries. In brief, I think you can indulge in an extended discussion that touches upon current thinking in these areas. Perhaps you could consider the following:

Major points

I would be interested to see your treatment of contemporary formulations in light of your focus on relative entropy. I am most familiar with the free energy principle (FEP); however, it would also be nice if you were able to briefly comment on the work by van sure (Vanchurin 2020, Vanchurin, Wolf et al. 2022, Vanchurin, Wolf et al. 2022). In addition, there are some ‘usual’ references that people will expect to see when noting that replicator dynamics and Bayesian belief updating provide apt explanations for biological information processing in evolution. For example, (Ao 2005, Sella and Hirsh 2005, Ao 2008, Frank 2012, Campbell 2016, Ramirez and Marshall 2017). And perhaps (Campbell 2021).

In relation to the FEP, there are a number of relevant publications that speak to your ideas. An early paper that focuses on the dynamics of non-equilibrium systems is (Friston and Ao 2012). Other papers that you might enjoy include: (Ramstead, Badcock et al. 2018, Ramstead, Sakthivadivel et al. 2022). More recently, there are some key developments that could be summarised along the following lines:

“Recent developments under the free energy principle converge upon the maximum entropy principle under constraints. In brief, it is now apparent that the FEP is dual to the constrained maximum entropy principle and — in its path integral formulation – a constrained maximum calibre principle (Sakthivadivel 2022, Sakthivadivel 2022). Put simply, one can regard a free energy functional as a relative entropy (i.e., KL divergence) that is augmented with a potential or energy term supplying constraints. The variational free energy in the FEP can be variously rearranged to be expressed as constraints provided by an expected surprisal (under a generative model or attracting set: c.f., q above) and the entropy of a variational density encoding knowledge or information about the external milieu. Another rearrangement has the same functional form as ACTION above; namely a decomposition into (the path integral of) a KL divergence and energy term; read here in terms of potential and kinetic information, respectively.

Related decompositions of variational free energy have the same form; for example, a decomposition into surprisal (a.k.a., self information or negative log evidence) and the KL divergence between posterior and prior densities (a.k.a., complexity or information gain). The decomposition of free energy in terms of self information is interesting from two perspectives. First, the action or path integral of self information is the entropy of a systems boundary states (e.g., sensations or information from the environment). This leads to the slightly paradoxical notion that the maximum entropy principle (Jaynes 1957) minimises the entropy of exchange with the environment. Variational free energy is used in machine learning to optimise generative models of data; illustrating the fact that minimising variational free energy is equivalent to maximising model evidence (a.k.a. marginal likelihood) or — in the context of self organisation — self evidencing (Hohwy 2016). On some readings, the marginal likelihood of a model or phenotype is read as adaptive fitness (Constant, Ramstead et al. 2018).”

I would be interested in hearing how you relate potential and kinetic information to variational free energy (or perhaps the expected free energy that is, effectively, the path integral or action of free energy).

Minor points

Line 174: here, you refer to informational communiques originating from within the “usual considered boundaries of the organism”. It would be useful at this point to introduce the notion of a Markov blanket that individuates internal from external states and provides an informational boundary; e.g., (Pellet and Elisseeff 2008, Clark 2017, Palacios, Razi et al. 2017, Parr, Da Costa et al. 2020). The Markov blanket is the central construct of the FEP that licences the inference perspective that you develop.

Line 190: you talk about a statistical manifold as a two-dimensional geometry which can represent the mean and variance of a probability distribution. As I’m sure you know, information geometries are not restricted to two-dimensional manifolds. However, they may be privileged on a number of counts. First, under the maximum entropy principle, the densities in question will be Gaussian and therefore the information geometry does indeed reduce to a two-dimensional statistical manifold. This becomes practically relevant in data analysis and statistics in several guises; for example, parametric statistics assumes random gaussian fluctuations (as does much of physics) resulting in what is sometimes referred to as the Laplace approximation. You might want to mention this?

Line 218: I recommend you change " … which moves continuously and irreversibly along the entropy gradient" to:

"Dissipative entropy dynamics then follow the trajectory of the system, which moves continuously and irreversibly along free energy gradients in a geodesic space of probable states."

Irrespective of a commitment to the FEP, it is a free energy that is minimised in open and closed systems. In other words, the free energy or partition function scores the divergence between the current density and the final (steady-state) density. Furthermore, the irreversible aspect is only relevant for systems that possess detailed balance and have an equilibrium steady-state. Work by Ping Ao (referenced above) places a special emphasis on divergence free dynamics seen in biotic self organisation that are definitive of a nonequilibrium steady-state; in which there is a mixture of time reversible (e.g., biorhythms and life cycles) and irreversible dynamics (e.g., dissipation and death). You can either elude this issue by referring explicitly to dissipative dynamics and ignoring solenoidal dynamics. Alternatively, you could indulge in a brief paragraph foregrounding the importance of conservative dynamics that break detailed balance; e.g., Red Queen dynamics (Zhang, Xu et al. 2012).

Line 223: I did not understand the sentence:

"The advantage of utilising this geometric approach is its capacity for a larger perspective and fundamental analysis, whether focus on the most probable trajectory is given by entropy and guided biological inference principles and not based on proposed laws of physics or any action principle."

This did not make any sense. Variational principles of least action are dual to the information geometry approach — in which the most probable trajectory is the path of least action. This means that one can move gracefully between least action principles, gauge theoretic treatments and information geometric treatments. They are all the same. For example, the FEP can be described as a principle of stationary action under the path integral formulation (Friston, Da Costa et al. 2022) or as an instance of a constrained maximum entropy principle (Sakthivadivel 2022). Both lend themselves to a gauge theoretic treatment (Sengupta, Tozzi et al. 2016) and both call upon information geometry, information length and information rates to describe the implicit belief updating (Da Costa, Parr et al. 2021). Perhaps you could unpack and qualify this sentence?

Line 260: I think there has been a font substitution error, where the rate of change of x has been replaced by X.

Line 275: when you say "the analysis of biological systems was structured" are you referring to some previous work or are you describing what you did to present the results of this paper?

Line 321: there is a missing subscript (i) on the rate of change of p.

Line 361: in most texts, there is a double bar in the expression for the KL divergence; i.e., D_KL[Q||P]

Line 392: could you be notationally explicit about what the integral is over? Is it over the support of the normalised or unnormalized probability densities or is it over time?

Line 449: I was intrigued by the notion that the Lyapunov vector signature of stability for these dynamics can also serve as a quantitative metric for biosemiotics.

This reminded me of the gauge theoretic treatment of the FEP. In brief, the Fisher information matrix becomes the curvature of variational free energy. This means that the Lyapunov exponents of any flow on that free energy landscape (i.e., statistical manifold) come to encode the curvature and thereby the precision or confidence in encoded beliefs. This also determines the information length associated with any movement on the manifold. On this view, it equips the statistical manifold with a metric that affords the information geometry. I would be interested in hearing the intuitions that led to this sentence (perhaps detailed in references 39 and 40)?

Line 466 and 473: I suspect that people familiar with the FEP will raise an eyebrow when you say that biological systems:

"deconstruct information about the bio-continuum as a primary whole rather than constructing representations” and, implicitly, "without the need for representational models".

The premise of the FEP is that representational (generative models) emerge from density dynamics. I think you are absolutely right to point out that the notion of a representation in a generative model is not the cause of the dynamics but only an interpretation of the dynamics. However, when it comes to simulating self organisation, one invariably starts with a generative model and then derives the dynamics as gradient flows on a free energy. In this sense, it can be applied in an epistemological sense. Having said this, the free energy principle itself is meant to be a method that allows you to interpret self organisation in terms of inference, measurement or observation. Note that the representational aspect of a generative model entails a whole – not parts: the only parts inherit from mean field approximations such as hierarchical structure implicit in generative models of the kind used in deep learning and by the brain.

Although these are somewhat philosophical issues – e.g., (Bruineberg, Dolega et al. 2021) — I think you could devote a paragraph to unpacking and nuancing the connection between density dynamics and the implicit representationalism that inherits from having a boundary or Markov blanket.

I hope that these comments and questions help should any revision be required.

Ao, P. (2005). "Laws in Darwinian evolutionary theory." Physics of Life Reviews 2(2): 117-156.

Ao, P. (2008). "Emerging of Stochastic Dynamical Equalities and Steady State Thermodynamics from Darwinian Dynamics." Commun Theor Phys 49(5): 1073-1090.

Bruineberg, J., K. Dolega, J. Dewhurst and M. Baltieri (2021). "The Emperor's New Markov Blankets." Behav Brain Sci: 1-63.

Campbell, J. O. (2016). "Universal Darwinism As a Process of Bayesian Inference." Front Syst Neurosci 10(49): 49.

Campbell, J. O. (2021). The Knowing Universe, Amazon Digital Services LLC - KDP Print US.

Clark, A. (2017). How to Knit Your Own Markov Blanket. Philosophy and Predictive Processing. T. K. Metzinger and W. Wiese. Frankfurt am Main, MIND Group.

Constant, A., M. J. D. Ramstead, S. P. L. Veissiere, J. O. Campbell and K. J. Friston (2018). "A variational approach to niche construction." J R Soc Interface 15(141).

Da Costa, L., T. Parr, B. Sengupta and K. Friston (2021). "Neural Dynamics under Active Inference: Plausibility and Efficiency of Information Processing." Entropy (Basel) 23(4): 454.

Frank, S. A. (2012). "Natural selection. V. How to read the fundamental equations of evolutionary change in terms of information theory." J Evol Biol 25(12): 2377-2396.

Friston, K. and P. Ao (2012). "Free energy, value, and attractors." Comput Math Methods Med 2012: 937860.

Friston, K., L. Da Costa, D. A. R. Sakthivadivel, C. Heins, G. A. Pavliotis, M. Ramstead and T. Parr (2022) "Path integrals, particular kinds, and strange things." arXiv:2210.12761.

Hohwy, J. (2016). "The Self-Evidencing Brain." Nous 50(2): 259-285.

Jaynes, E. T. (1957). "Information Theory and Statistical Mechanics." Physical Review Series II 106(4): 620–630.

Palacios, E. R., A. Razi, T. Parr, M. Kirchhoff and K. Friston (2017). "Biological Self-organisation and Markov blankets." bioRxiv.

Parr, T., L. Da Costa and K. Friston (2020). "Markov blankets, information geometry and stochastic thermodynamics." Philos Trans A Math Phys Eng Sci 378(2164): 20190159.

Pellet, J. P. and A. Elisseeff (2008). "Using Markov blankets for causal structure learning." Journal of Machine Learning Research 9: 1295-1342.

Ramirez, J. C. and J. A. R. Marshall (2017). "Can natural selection encode Bayesian priors?" J Theor Biol 426: 57-66.

Ramstead, M. J. D., P. B. Badcock and K. J. Friston (2018). "Answering Schrodinger's question: A free-energy formulation." Phys Life Rev 24: 1-16.

Ramstead, M. J. D., D. A. R. Sakthivadivel, C. Heins, M. Koudahl, B. Millidge, L. Da Costa, B. Klein and K. J. Friston (2022) "On Bayesian Mechanics: A Physics of and by Beliefs." arXiv:2205.11543.

Sakthivadivel, D. A. R. (2022) "A Constraint Geometry for Inference and Integration." arXiv:2203.08119.

Sakthivadivel, D. A. R. (2022) "Towards a Geometry and Analysis for Bayesian Mechanics." arXiv:2204.11900.

Sella, G. and A. E. Hirsh (2005). "The application of statistical physics to evolutionary biology." Proc Natl Acad Sci U S A 102(27): 9541-9546.

Sengupta, B., A. Tozzi, G. K. Cooray, P. K. Douglas and K. J. Friston (2016). "Towards a Neuronal Gauge Theory." PLoS Biol 14(3): e1002400.

Vanchurin, V. (2020). "The World as a Neural Network." Entropy 22(11): 1210.

Vanchurin, V., Y. I. Wolf, M. I. Katsnelson and E. V. Koonin (2022). "Toward a theory of evolution as multilevel learning." Proc Natl Acad Sci U S A 119(6): e2120037119.

Vanchurin, V., Y. I. Wolf, E. V. Koonin and M. I. Katsnelson (2022). "Thermodynamics of evolution and the origin of life." Proc Natl Acad Sci U S A 119(6): e2120042119.

Zhang, F., L. Xu, K. Zhang, E. Wang and J. Wang (2012). "The potential and flux landscape theory of evolution." J Chem Phys 137(6): 065102.

Author Response

The comments from this reviewer were excellent and reflects his expertise on the subject matter.

Major Comments:

Many of the suggested references have been added with the appropriate context.

The possible relationship of the potential and kinetic differential to variational free energy has been included. 

Minor Comments:

Most of the specific line-item comments have been addressed including 218, 260, 275, 321, 365,392.  Those not explicitly addressed were considered beyond the scope and intent of this special issue.

The Lyapunov signature biosemiotics is in reference to a prior paper of mine that is now included in the references and uploaded for your purview.  

The difference between deconstruction and representational is somewhat philosophical and I have spent a great deal of time and thought in the past on this issue.  It is an important distinction that I am firm on but consider the background arguments as beyond the scope of this paper. See below for some ideas. 

Edmund Husserl thought that our perceptions do not represent an object but rather directly presents the object. Furthermore, Huw Price contends that our thoughts and mental constructs are not simply representations of something external and objectively perceived as they are often derived internally and therefore must incorporate some degree of a subjective characteristic.  The well-known American philosopher Richard Rorty also rejected the notion that the construction of internal representations of objects of perceived reality were necessary for the creation of knowledge. Rorty contended that the development of knowledge requires temporal and historical context.  Following this perspective, it could be concluded that meaning plays a critical role in the development of knowledge that is not typically captured in representational models.  Rovelli further argues that the role of Science as Natural Philosophy should be about more than just ostensible informational representations as a predictive description of phenomena.

Reviewer 3 Report

attachment

Author Response

There are more references from contributing literature now added. 

The derivation of the final equations and mathematical platform are naturally derived from historic foundations, but I assure the reviewer the context and application of the work is original. 

A practical application is demonstrated in my recent publication in Biosemiotics and uploaded for your purview.

The other comments are beyond the scope and intent of the current special issue for which this was submitted.  

Round 2

Reviewer 1 Report

I asked the author to add an example of applying the proposed idea to a specific physical system. Providing a concrete example would improve the paper in that it would strongly support the validity of your ideas. I hope the next follow-up will pay more attention to this.

Reviewer 2 Report

Many thanks for attending to my previous suggestions.

Reviewer 3 Report

The author has addressed most of my comments